# Increasing coverage in cervical and colorectal cancer screening by leveraging attendance at breast cancer screening: A cluster-randomised, crossover trial

**Anne Dorte Lerche Helgestad**[1,2]*, **Mette Bach Larsen**[1,2,3], **Sisse Njor**[2,4], **Mette Tranberg**[1,2,5], **Lone Kjeld Petersen**[1,6,7], **Berit Andersen**[1,2]

1 University Research Clinic for Cancer Screening, Department of Public Health Programmes, Randers Regional Hospital, Randers, Denmark, 2 Department of Clinical Medicine, Aarhus University, Aarhus, Denmark, 3 Research Unit, Horsens Regional Hospital, Horsens, Denmark, 4 Department of Data, Innovation and Research, Lillebaelt Hospital, Vejle; University Hospital of Southern Denmark, Odense, Denmark, 5 Department of Pathology, Randers Regional Hospital, Randers, Denmark, 6 Department of Obstetrics and Gynaecology, Odense University Hospital, Odense, Denmark, 7 Department of Clinical Medicine, University of Southern Denmark, Odense, Denmark

* annesper@rm.dk

## Abstract

### Background

Screening participation remains suboptimal in cervical cancer (CC) and colorectal cancer (CRC) screening despite their effectiveness in reducing cancer-related morbidity and mortality. We investigated the effectiveness of an intervention by leveraging the high participation rate in breast cancer (BC) screening as an opportunity to offer self-sampling kits to nonparticipants in CC and CRC screening.

### Methods and findings

A pragmatic, unblinded, cluster-randomised, multiple period, crossover trial was conducted in 5 BC screening units in the Central Denmark Region (CDR) between September 1, 2021 and May 25, 2022. On each of 100 selected weekdays, 1 BC screening unit was randomly allocated as the intervention unit while the remaining units served as controls. Women aged 50 to 69 years attending BC screening at the intervention unit were offered administrative check-up on their CC screening status (ages 50 to 64 years) and CRC screening status (aged 50 to 69), and women with overdue screening were offered self-sampling. Women in the control group received only standard screening offers according to the organised programmes. The primary outcomes were differences between the intervention group and the control group in the total screening coverage for the 2 programmes and in screening participation among women with overdue screening, measured 6 months after the intervention. These were assessed using intention-to-treat analysis, reporting risk differences with 95% confidence intervals (CIs).

**Data Availability Statement:** Under Danish law, restrictions apply to the availability of the data generated during the study. Register data were

PLOS MEDICINE

Leveraging breast cancer screening to increase cervical and colorectal cancer screening coverage

used under a license for the present study and may be available upon reasonable request to the Danish Health Data Authority (kontakt@sundhedsdata.dk) and Statistics Denmark (dst@dst.dk). The participants were not asked to give consent for publication of the questionnaire data.

**Funding:** The author(s) received no specific funding for this work.

**Competing interests:** Roche Diagnostics sponsored the Cobas 4800 HPV DNA tests in this study. According to the contract between Roche Diagnostics and the University Research Clinic for Cancer Screening, Randers Regional Hospital, Denmark, Roche Diagnostics had no influence on the scientific process and no editorial rights pertaining to this manuscript. MT, LKP and BA have participated in other studies with human papillomavirus (HPV) DNA tests sponsored by Roche Diagnostics. MT has received honoraria from Roche Diagnostics for lectures on HPV self-sampling.

**Abbreviations:** BC, breast cancer; CC, cervical cancer; CDR, Central Denmark Region; CI, confidence interval; CPR, civil personal registration; CRC, colorectal cancer; FIT, faecal immunochemical test; GP, general practitioner; HPV, human papillomavirus; RD, risk difference.

A total of 27,116 women were included in the trial, with 5,618 (20.7%) in the intervention group and 21,498 (79.3%) in the control group. Six months after the intervention, total coverage was higher in the intervention group as compared with the control group in CC screening (88.3 versus 83.5, difference 4.8 percentage points, 95% CI [3.6, 6.0]; $p < 0.001$) and in CRC screening (79.8 versus 76.0, difference 3.8 percentage points, 95% CI [2.6, 5.1]; $p < 0.001$). Among women overdue with CC screening, participation in the intervention group was 32.0% compared with 6.1% in the control group (difference 25.8 percentage points, 95% CI [22.0, 29.6]; $p < 0.001$). In CRC screening, participation among women overdue with screening in the intervention group was 23.8% compared with 8.9% in the control group (difference 14.9 percentage points, 95% CI [12.3, 17.5]; $p < 0.001$). Women who did not participate in BC screening were not included in this study.

## Conclusions

Offering self-sampling to women overdue with CC and CRC screening when they attend BC screening was a feasible intervention, resulting in an increase in participation and total coverage. Other interventions are required to reach women who are not participating in BC screening.

## Trial registration

ClinicalTrials.gov NCT05022511. The record of processing activities for research projects in the Central Denmark Region (R. No.: 1-16-02-217-21).

## Author summary

### Why was this study done?

- Participation rates in cervical cancer (CC) and colorectal cancer (CRC) screening lag significantly behind those of breast cancer (BC) screening in Denmark, despite the proven preventive benefits in reducing cancer-related morbidity and mortality.

- Previous research has identified numerous barriers to nonparticipation across different screening programmes, prompting suggestions for combined screening services as a potential solution.

- Considering that women attending BC screening fall within the age range eligible for CC and CRC screening, leveraging the BC screening visit could serve as an opportune moment to engage in the other screening programmes.

### What did the researchers do and find?

- We conducted a randomised controlled trial including 27,116 women aged 50 to 69 years attending BC screening, of whom 5,618 women received an intervention offering self-sampled CC and/or CRC screening if overdue for screening and within the eligible screening age range.

PLOS Medicine | https://doi.org/10.1371/journal.pmed.1004431   August 13, 2024

2 / 19

- The intervention proved effective in increasing coverage with 4.8 percentage points in CC screening and 3.8 percentage points in CRC screening compared to the control group.

- The intervention was feasible and well received by the women, without affecting their high satisfaction with BC screening.

### What do these findings mean?

- This study suggests that combining preventive services may offer a pragmatic strategy to enhancing the effectiveness of public health interventions in various settings.

- The intervention strategy can be easily expanded and could increase screening participation if implemented on a larger scale.

- The main limitation of the study is that the intervention focuses solely on women attending BC screening, and a different approach is required for women not participating in this screening.

## Introduction

Screening for breast cancer (BC), cervical cancer (CC), and colorectal cancer (CRC) is recommended because these programmes reduce cancer-related morbidity [1] and mortality [2–4]. For screening programmes to be effective, participation rates must be high. Nevertheless, many cancer screening programmes suffer from suboptimal participation rates [5]. Reasons for nonparticipation have been studied extensively, and conclusions vary according to programme structures [6,7]. Among numerous barriers, one reason mentioned across screening programmes is temporary impediments or forgetfulness. This indicates that some nonparticipants do not make a deliberate choice not to participate but unwittingly delay and forget to take part in screening [7–10]. Therefore, various interventions to minimise logistic barriers have been conducted, such as offering self-sampled CRC screening with a stool test [11] and vaginal self-sampling kits for high-risk human papillomavirus (HPV) testing in CC screening [11–13]. For the subgroup that tends to forget screening offers, a concept offering multiple screenings at the same time has been suggested as a potential method to enhance screening participation [14,15]. Interventions at the relational level, such as endorsement from a healthcare person and more personalised reminders, have also proven effective in enhancing participation [7,11,12]. However, these interventions may often be resource-intensive [12].

Participating in one screening programme increases the probability of participating in other screening programmes [16–18]. While the participation rate in BC screening in Denmark has consistently remained high (above 80% as of 2023) [19], the participation rates of both CC and CRC screening remain around 60% [20], resulting in a participation gap in these 2 screening programmes [18], despite all programmes being offered free of charge. The same picture is seen in other western countries with organised screening programmes and comparable healthcare services [14,21,22].

In Denmark, screening invitations are sent directly to residents within the eligible age range whenever they are due. Women aged 50 to 69 years receive a prescheduled appointment

for BC screening, while women aged 23 to 64 years are invited to book appointments for CC screening. Both women and men aged 50 to 74 years receive self-sampling kits for CRC screening. This approach results in a cohort of women who are eligible for all 3 screening programmes at the same time.

Therefore, BC screening may serve as a valuable opportunity to motivate and inform about the other 2 screening programmes, as also proposed by others [10,15,21,23,24]. However, to our knowledge, the effects of doing so have not been subjected to clinical study.

The aim of this trial was to evaluate if BC screening attendance represents a significant window of opportunity for offering self-sampled CC and CRC screening to women who are overdue for CC and CRC screening, and hereby increase coverage in the targeted population.

## Methods

### Setting

In Denmark, health care is organised into 5 administrative regions. The trial was conducted in the Central Denmark Region (CDR), which is inhabited by approximately 1.3 million residents, corresponding to 22% of the total Danish population.

Cancer screening is decided nationally and administered regionally. BC screening with mammography is offered biennially to women aged 50 to 69 years who receive digital invitations to attend a prebooked appointment at a regional unit. A reminder is sent in case of nonattendance. In the CDR, 5 BC screening units serve women for BC screening 5 days a week. CC screening is offered to women aged 23 to 64 years. Women aged 50 to 64 years are offered screening every 5 years, whereas women below 50 years are invited every 3 years. Women receive digital invitations to undergo cervical cytology sampling during a pelvic examination with a general practitioner (GP). Liquid-based cytology samples are analysed for abnormal cells and/or tested for high-risk HPV types [20]. If a cervical sample is not registered after an invitation, up to 2 reminders are sent at 3- and 6-month intervals. Throughout the trial, self-sampling was not offered as part of the national CC screening programme. CRC screening is offered biennially to all residents aged 50 to 74 years with a faecal immunochemical test (FIT) taken as a self-sample and mailed by post. The samples are analysed for haemoglobin with a cutoff value of 100 ng haemoglobin (HB)/ml buffer. A digital reminder is sent if a faecal sample is not registered within 45 days of the invitation. All 3 screening programmes operate on integrated call-recall systems, where administrative systems automatically keep track of when a woman is to receive an invitation, a reminder or if she has unsubscribed. In cases of opportunistic screening, invitations are deferred to align with the recommended screening intervals. Healthcare is tax-funded and cancer screening, including any follow-up and treatment, is offered free of charge to all residents in the screening-eligible age.

All Danish residents have a unique 10-digit civil personal registration (CPR) number. The CPR number is used for any contact with the Danish healthcare system and allows linkage of personal data from the Danish healthcare registries [25].

### Study population

The study population included all women attending BC screening in the CDR on the intervention days. For the CC screening intervention, only women aged 50 to 64 years were included as women above that age were no longer in the target population for this screening programme. For the CRC screening intervention, women were included only if they had received their first CRC screening invitation more than 4.5 months before their BC screening. Individuals attending BC screening during the intervention period were included irrespective of their gender.

The exclusion criteria included a prior diagnosis of CC/CRC and prior hysterectomy (only in the CC screening). Women who were lost to follow-up (death or emigration within 6 months after the intervention) were not included in the analyses.

## Study design

The rationale and design of this trial have been described in detail elsewhere [26].

The study was designed as a pragmatic, unblinded, multiple period, cluster-randomised, crossover trial [27] conducted with 100 cluster periods in the 5 BC screening units located in the CDR. The study was conducted from September 1, 2021 to May 25, 2022, with follow-up until November 25, 2022.

A randomisation scheme was employed to ensure balanced allocation of BC screening units throughout the study period. For each selected intervention day, one of the 5 screening units was randomly assigned to host the intervention, while the remaining 4 units served as controls, corresponding to a 1:4 allocation ratio. Accordingly, all BC screening units had 20 days of intervention and 80 days serving as controls. This approach was adopted to avoid the potential introduction of systematic differences between the intervention and control groups due to variations in the populations associated with each screening unit.

The randomisation of screening units was performed by an independent data manager using a computer-generated random number sequence in the statistical software STATA V.16. Women attending BC screening on the intervention days remained unaware of the randomisation. However, blinding of participants and study staff was unfeasible given the nature of the intervention.

The study is reported as per the Consolidated Standards of Reporting Trials (CONSORT) guideline for randomised trials (S1 Checklist).

## Intervention

At the intervention unit, a research assistant asked women if they were interested in having a check-up on their last day of participation in CC and/or CRC screening. The research assistant used the administrative systems to determine the screening status if consent was obtained. Women eligible for CC screening but overdue for screening were offered to receive a vaginal brush (Evalyn Brush, Rovers Medical Devices, the Netherlands [28]) for HPV self-sampling. Women overdue for CRC screening were offered to receive a FIT (OC Sensor System, Eiken Chemical Company, Japan) for blood trace detection. Both screening kits were forwarded by mail after the intervention day along with screening programme information and picture-based information material showing how to use the self-sampling kit.

Women were considered overdue for CC screening if they had never participated, if they had no record of a cervical sample within the past 5 years and 6 months, or if they were non-responders to a screening invitation received more than 6 months ago. Women were considered overdue for CRC screening if they had no record of a FIT in the past 2 years and 4.5 months, or if they had not responded to an invitation received more than 4.5 months ago. The difference in time intervals reflects the fact that reminders are sent at different intervals in the 2 screening programmes. Thus, a woman was considered overdue in both screening programmes, if she had not responded to an invitation and 1 reminder within 3 months after the reminder was sent.

Women in the control group received standard screening offers, encompassing invitations and reminders for the other 2 screening programmes independently when due.

To ensure intervention fidelity, the intervention was alternately led by 2 research assistants, who received close supervision from the principal investigator, including reporting after each

day of interventions, weekly meetings to address questions, and regular on-site supervision during the intervention. A systematic schedule for categorising screening status was followed to ensure consistency in the approach.

## Follow-up and clinical management

Follow-up and clinical management were embedded into the Danish routine cancer screening programmes and national guidelines were followed. Consequently, vaginal self-samples were analysed for high-risk HPV (HPV16, HPV18, and 12 other high-risk types pooled together; HPV 31, 33, 35, 39, 45, 51, 52, 56, 58, 59, 66, 68) using the Cobas 4800 HPV DNA test [29]. The analyses were conducted as part of routine and validated laboratory procedures at the Department of Pathology at Randers Regional Hospital [30]. Women who tested positive for HPV through self-sampling were advised to schedule an appointment with their GP within 1 month for cervical cytology sample collection for triage purposes [31]. The triage process involved co-testing for HPV and cytology, which guided further clinical follow-up [32]. FIT samples were analysed for the presence of blood traces using a cutoff value of 100 ng haemoglobin (HB)/ml buffer at the Department of Clinical Biochemistry at Randers Regional Hospital, following national protocols [33]. Women with a positive FIT were booked for a colonoscopy within 14 days, and further clinical management was in accordance with national guidelines.

## Outcomes

The 2 primary outcomes were [26] the following:

1. Difference in total CC and/or CRC screening coverage between the intervention and control groups 6 months after the intervention. Assessed as the proportion of women adhering to CC screening for the past 3.5/5.5 years (depending on the woman's age) and/or adhering to CRC screening for the past 2 years and 4.5 months.

2. Difference in CC and/or CRC screening participation 6 months after the intervention between the intervention and control groups among women overdue for CC/CRC screening at the intervention date.

For CC screening, both self-sampled screening and GP-collected screening samples were included.

Secondary outcomes for both CC and CRC screening comprised: prevalence of positive self-samples, compliance with follow-up (CC: a GP-collected cervical sample within 180 days after a positive screen, CRC: colonoscopy within 60 days from a positive screen, both aligning with the Danish quality measures for timely follow-up), screening history of self-samplers (CC: "under-screened" defined as screened at least once within the 10 years leading up to the inclusion date but overdue at baseline, "unscreened" defined as no CC screening within the past 10 years; CRC: "under-screened" defined as a minimum of 1 FIT, but no FIT within the past 2 years and 4.5 months, and "unscreened" defined as no previous FIT despite invitation).

In CC screening, the response rate to the re-test offer and the incidence of HPV–positive cases after 12 months were calculated for women aged 60 to 64 years with an initial HPV–negative self-sample.

In pursuance of Danish data protection legislation, further clinical follow-up, as outlined in the protocol [26] (histology after biopsies obtained at colposcopies/colonoscopies and referral rate for colposcopy after an HPV–positive CC screening), is not reported due to the small numbers of cases.

Post-protocol, secondary outcomes also included the proportion of women eligible for all 3 programmes, the proportion of these women overdue for both CC and CRC screening, and

their participation in both screenings within the follow-up period to describe participation patterns for women eligible in all 3 programmes and overdue for both CC and CRC screening.

Process outcomes comprised the proportion of women consenting to a check-up on their CC and CRC screening status, the proportion overdue for CC and CRC screening, the proportion accepting to receive a self-sampling kit, and the proportion not returning the self-sample. Finally, we assessed satisfaction with BC screening to examine potential adverse effects related to the BC screening experience due to the intervention. Additionally, the acceptability of the intervention was evaluated to gauge the quality of the delivered intervention.

## Data sources

The CPR number was used to link data across the registries and to obtain current personal data for both the intervention and control groups. Data collected for main, secondary, and process outcomes are described in details elsewhere [26] and listed in Table 1.

To evaluate the acceptability of the intervention, a questionnaire (S1 Questionnaire) was mailed to all women in the intervention and the control group enquiring about their satisfaction with BC screening. The questionnaire was mailed within a week after they had attended BC screening. The questionnaire contained 5 questions adapted from the national investigation of patient experiences [34]. The questions covered the reception in the units, personnel's professionalism, trust in the examination, overall satisfaction with BC screening, and intention to participate in the next screening round. Women in the intervention group received 6 additional questions related to their experience with the intervention. These questions covered the oral and written information provided about the intervention, whether the overall information about the intervention was sufficient, meaningfulness for the individual, meaningfulness of a one-stop offer for all 3 cancer screenings, and the participants' intention to accept a similar offer another time. Questionnaire responses were given on a Likert scale ranging from 1 to 4 ("1" being the best possible) or "Do not know."

**Table 1. Registers and data collected.**

| Register | Data collected |
|---|---|
| The regional administrative system of the Breast Cancer Screening Programme | Identifying the study population |
| Invitation and Administration Module (IAM) | CRC screening status* |
| The Danish Pathology Register (DPR) | CC screening status*<br>CC screening status and history**<br>Cervical cytology samples<br>HPV tests |
| The Danish CRC Screening Database | CRC screening status and history** |
| The Register of Laboratory Results for Research | FIT results |
| The Danish Cancer Registry (DCR) | Previous cancer diagnosis |
| The Danish National Patient Register (DNPR) | Hysterectomy<br>Colonoscopies |
| Statistics Denmark | Sociodemographic data (ethnicity, marital status, education) |
| The Danish Civil Registration System (CPR) | Vital status, migration |

*Utilised by the research assistants during the intervention, upon consent, to identify women overdue for screening.

**Retrieved at the end of follow-up period to obtain screening history before the intervention and the screening status on the day of the intervention.

CC, cervical cancer; CRC, colorectal cancer; FIT, faecal immunochemical test; HPV, human papillomavirus.

## Sample size

The expected attendance was 55 women a day in each BC screening unit. Among these, 40 women were expected to be eligible for CC screening and 52 women were expected to be eligible for CRC screening. Women not attending their appointment were excluded before data on the study population were collected. Accordingly, the CC screening study population was expected to comprise 4,000 women and the CRC screening study population 5,200 women within the chosen 100-day time frame. Based on preliminary data from a recent study on Danish women's concurrent participation in the cancer screening programmes [18], the assumption was that 20% of the women attending BC screening were nonparticipants in CC screening, and 35% of the women attending BC screening were nonparticipants in CRC screening. Hence, the trial had 90% power to detect a screening coverage difference of 2.3% in CC screening (increasing from 80.0% to 82.3%) and 2.4% in CRC screening (increasing from 65.0% to 67.4%). In this calculation, the potential effect of the cluster-randomised, crossover design was not considered, as the clusters served equally as intervention and control groups.

## Statistical analyses

Baseline characteristics of the intervention and control groups were described using frequency and percentages (%) for categorical variables, and median and interquartile range for age. Differences between the 2 groups were tested with chi-squared tests for categorical variables.

Coverage and participation were analysed in intention-to-treat analyses and estimated as risk differences (RDs) with 95% confidence intervals (CIs). Secondary outcomes and process outcomes were estimated with descriptive statistics with 95% CIs. The individuals within the clusters were considered independent of each other, and the intervention was assigned equally between the screening units during the study period. Hence, a design effect due to cluster randomisation was not considered in the main analyses. However, a sensitivity analysis of the primary outcomes was conducted using logistic regression with cluster-robust standard errors to account for clustering within screening units.

Questionnaire data were analysed by applying chi-squared tests to each question to determine satisfaction with BC screening, comparing "1–2" (positive) to "3–4" (negative). Questionnaire responses regarding the evaluation of the intervention were reported as frequencies and percentages of a positive opinion (answering "1" or "2").

A two-sided value of $p < 0.05$ was considered statistically significant. All analyses were performed using STATA V.17.

## Ethics statement

The study was listed in the record of processing activities for research projects in the Central Denmark Region (R. No.: 1-16-02-217-21). In pursuance of the Danish Consolidation Act on Research Ethics Review of Health Research Projects, the study was not notifiable to the Committee (R. No.: 1-10-72-1-21). No formal study protocol or informed consent was therefore required.

## Results

### Study population

A total of 27,116 women attended BC screening in the CDR on the intervention days; 5,618 (20.7%) women attending an intervention unit and 21,498 (79.3%) a control unit. Flow charts depicting the trial are presented in Figs 1 and 2.

At baseline, age, sociodemographic factors, and screening history were comparable between intervention and control groups in both screening programmes (Table 2). Among the total

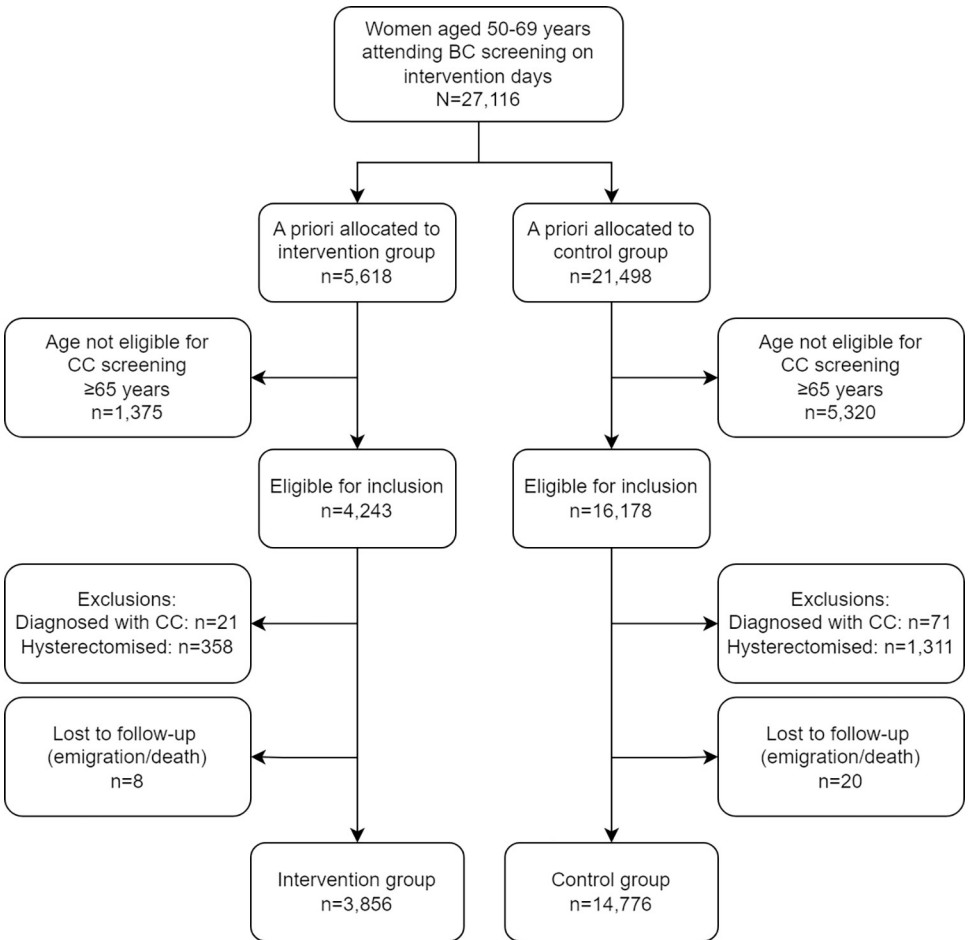

**Fig 1. Flow diagram of the study population in cervical cancer screening.** BC, breast cancer; CC, cervical cancer.

study population, 15,637 (57.7%) women were eligible for participation in both screening programmes, with 3,244 women in the intervention group and 12,393 in the control group. Within this subset, 8.6% ($n = 279$) in the intervention group and 8.2% ($n = 1,010$) in the control group were overdue in both screening programmes at baseline.

## Coverage and participation

For CC screening, baseline coverage was comparable between the intervention and the control group (Table 3). At 6 months after the intervention, coverage in the intervention group had increased to 88.3% (95% CI [87.3%, 89.3%]), corresponding to an increase of 4.8 percentage points (95% CI [3.6, 6.0]; $p < 0.001$) compared with the control group (Table 3). The increase was driven by a 32.0% (95% CI [28.3%, 35.8%]) screening participation of the women with overdue CC screening in the intervention group compared with 6.1% (95% CI [5.2%, 7.2%]) in the control group (RD = 25.8 percentage points, 95% CI [22.0, 29.6]; $p < 0.001$) (Table 4). An increase was detected for both under-screened women (RD = 26.9 percentage points, 95% CI [21.8, 32.0]; $p < 0.001$) and unscreened women (RD = 24.1 percentage points, 95% CI [18.4, 30.0]; $p < 0.001$) compared with the control group. Unscreened women participated statistically significantly more than under-screened women ($p = 0.02$). A total of 163 women (83.6%, 95% CI [77.6%, 88.5%]) participated with a self-sampled screening.

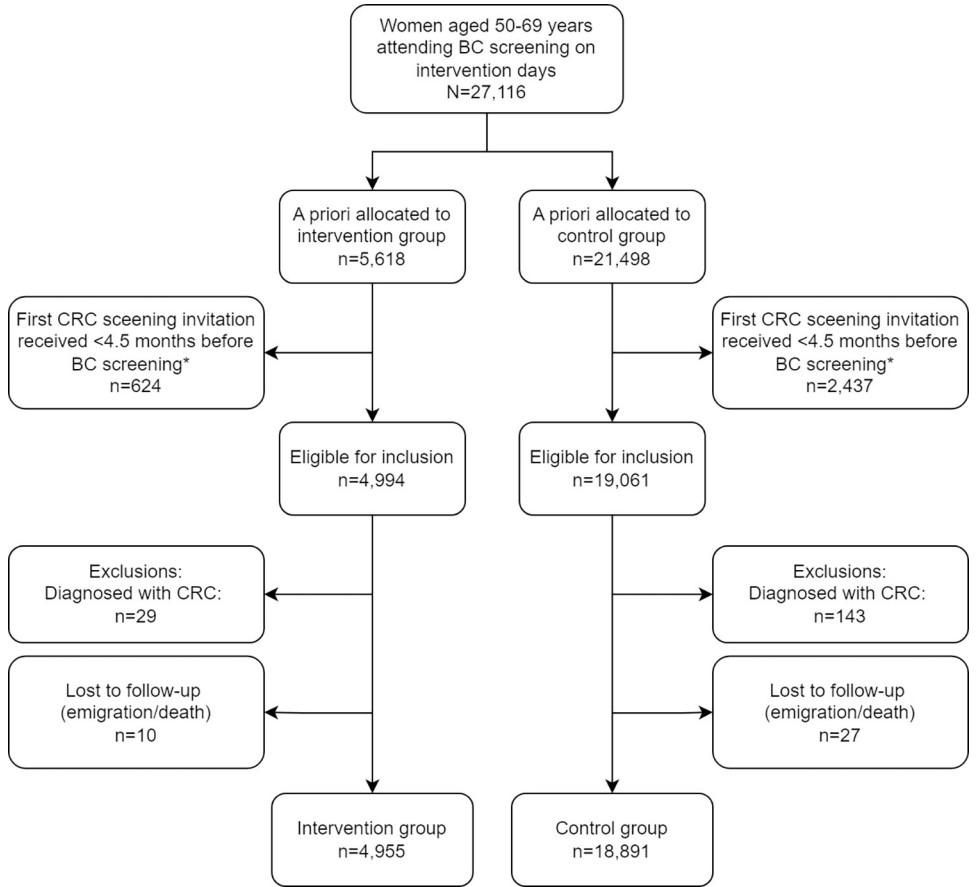

**Fig 2. Flow diagram of the study population in colorectal cancer screening.** BC, breast cancer; CRC, colorectal cancer.

For CRC screening, baseline coverage was comparable between the intervention and the control group (Table 3). Six months after the intervention, coverage in the intervention group had increased to 79.8% (95% CI [78.7%, 80.9%]), corresponding to an increase of 3.8 percentage points (95% CI [2.6, 5.1]; $p < 0.001$) compared with the control group. Among the women with overdue CRC screening, 23.8% (95% CI [21.4%, 26.3%]) participated in screening in the intervention group compared with 8.9% (95% CI [8.0%, 9.7%]) in the control group (RD = 14.9 percentage points, 95% CI [12.3, 17.5]; $p < 0.001$) (Table 4). The increase was observed in both under-screened women (RD = 18.9% percentage points, 95% CI [14.6, 23.2]; $p < 0.001$) and unscreened women (RD = 12.2 percentage points, 95% CI [9.1, 15.3]; $p < 0.001$). No statistically significant difference in participation was found between under-screened and unscreened women ($p = 0.81$).

The sensitivity analysis of the primary outcomes, accounting for clustering within the screening units, did not alter the estimates (S1 Table).

## Secondary outcomes

For those participating in the CC screening intervention with a vaginal self-sample, the HPV DNA prevalence was 8.0% ($n/N = 13/163$; 95% CI [4.3%, 13.3%]). The follow-up compliance at the GP after an HPV–positive self-sample was 100% within 180 days, and the majority was compliant within the 30-day recommendation in the trial (exact numbers cannot be presented

**Table 2. Baseline characteristics of the study population.**

| | CC screening | | CRC screening | |
|---|---|---|---|---|
| | Intervention group $N$ = 3,856 $n$ (%) | Control group $N$ = 14,776 $n$ (%) | Intervention group $N$ = 4,955 $n$ (%) | Control group $N$ = 18,891 $n$ (%) |
| Age (years) | | | | |
| 50–55 | 1,713 (44.4%) | 6,425 (43.5%) | 1,179 (23.8%) | 4,361 (23.1%) |
| 56–60 | 1,177 (30.5%) | 4,699 (31.8%) | 1,297 (26.2%) | 5,141 (27.2%) |
| 61–65 | 966 (25.1%) | 3,652 (24.7%) | 1,384 (27.9%) | 5,148 (27.3%) |
| 66–70 | 0 (0%) | 0 (0%) | 1,095 (22.1%) | 4,241 (22.5%) |
| Ethnicity** | | | | |
| Danish | 3,546 (92.0%) | 13,679 (92.6%) | >4,637* (93.6%) | 17,752 (94.0%) |
| Western migrants | 87 (2.3%) | 340 (2.3%) | 95 (1.9%) | 383 (2.0%) |
| Non-western migrants | 218 (5.7%) | 735 (5.0%) | 216 (4.4%) | 751 (4.0%) |
| Missing | 5 (0.1%) | 22 (0.2%) | <5* (0.1%) | 5 (0.0%) |
| Marital status | | | | |
| Cohabitating | 2,878 (74.6%) | 10,963 (74.2%) | >3,670* (74.1%) | 13,897 (73.6%) |
| Living alone | 973 (25.2%) | 3,791 (25.7%) | 1,278 (25.8%) | 4,989 (26.4%) |
| Missing | 5 (0.1%) | 22 (0.2%) | <5* (0.1%) | 5 (0.0%) |
| Education | | | | |
| Low (≤10 years) | 516 (13.4%) | 2,190 (14.8%) | 937 (18.9%) | 3,603 (19.1%) |
| Middle (11–15 years) | 1,960 (50.8%) | 7,457 (50.5%) | 2,330 (47.0%) | 8,983 (47.6%) |
| High (>15 years) | 1,334 (34.6%) | 4,983 (33.7%) | 1,638 (33.1%) | 6,123 (32.4%) |
| Missing | 46 (1.2%) | 146 (1.0%) | 50 (1.0%) | 182 (1.0%) |
| Screening history** | | | | |
| Unscreened | 247 (6.4%) | 1,036 (7.0%) | 698 (14.1%) | 2,610 (13.8%) |
| Under-screened | 363 (9.4%) | 1,388 (9.4%) | 484 (9.8%) | 1,856 (9.8%) |
| Timely screened | 3,246 (84.2%) | 12,352 (83.6%) | 3,773 (76.1%) | 14,425 (76.4%) |

*Exact numbers are not given in pursuance of Danish data protection legislation, which does not permit reporting of individual data where total numbers are lower than 3.

**Western migrants: individuals originating from countries within the European Union, Andorra, Australia, Canada, Iceland, Liechtenstein, Monaco, New Zealand, Norway, San Marino, Switzerland, and the United States of America. Non-western migrants: originating from all other countries [26].

***Cervical cancer screening: unscreened was defined as no cervical sample within the past 10 years; under-screened was defined as screened at least once within the past 10 years but overdue at baseline; timely screened was defined as screened within the past 3.5/5.5 years according to the age of the woman at her last screening. Colorectal cancer screening: unscreened was defined as no previous FIT despite an invitation; under-screened was defined as a minimum of 1 FIT but overdue at baseline; timely screened was defined as screened within the past 2 years and 4.5 months.

CC, cervical cancer; CRC, colorectal cancer; FIT, faecal immunochemical test.

in pursuance of Danish data protection legislation). Of the women aged 60 to 64 years with an initial negative HPV self-sample, a total of 75.6% ($n/N$ = 34/45; 95% CI [60.4, 87.1]) completed the self-sample re-test sent 12 months later. All women who complete the re-test were HPV–negative.

Among the women with overdue CRC screening at the intervention units, the prevalence of positive FIT cases within the next 6 months was 2.5% ($n/N$ = 7/281; 95% CI [1.0%, 5.1%]). Among these, the majority had a colonoscopy within the recommended 60 days after a positive screening (exact numbers cannot be presented in pursuance of Danish data protection legislation).

For eligible women overdue in both screening programmes, 13.6% ($n/N$ = 38/279, 95% CI [9.8%, 18.2%]) in the intervention group participated in both screenings during follow-up, while 18.3% ($n/N$ = 51/279, 95% CI [13.9%, 23.3%]) participated in only 1 screening. In comparison, 0.3% ($n/N$ = 3/1,010, 95% CI [0.06%, 0.9%]) in the control group took part in both screenings, and 9.4% ($n/N$ = 95/1,010, 95% CI [7.7%, 11.4%]) participated in one.

**Table 3. Coverage in CC and CRC screening at baseline and after 6 months.**

| | Intervention group<br>*n/N*<br>%<br>[95% CI] | Control group<br>*n/N*<br>%<br>[95% CI] | RD<br>Percentage points<br>[95% CI] | *P*-value* |
|---|---|---|---|---|
| CC screening | | | | |
| Baseline | 3,246/3,856<br>84.2%<br>[83.0%, 85.3%] | 12,352/14,776<br>83.6%<br>[83.0%, 84.2%] | 0.6<br>[−0.7, 1.9] | |
| Six months after intervention | 3,406/3,856<br>88.3%<br>[87.3%, 89.3%] | 12,343/14,776<br>83.5%<br>[82.9%, 84.1%] | 4.8<br>[3.6, 6.0] | <0.001 |
| CRC screening | | | | |
| Baseline | 3,773/4,955<br>76.1%<br>[74.9%, 77.3%] | 14,425/18,891<br>76.4%<br>[75.7%, 77.0%] | −0.2<br>[−1.5, 1.1] | |
| Six months after intervention | 3,955/4,955<br>79.8%<br>[78.7%, 80.9%] | 14,356/18,891<br>76.0%<br>[75.4%, 76.6%] | 3.8<br>[2.6, 5.1] | <0.001 |

*n/N*, number of women screened/total number of women included in the group.

*Comparison between the intervention and control groups, tested using a chi-squared test.

95% CI, 95% confidence interval; CC, cervical cancer; CRC, colorectal cancer; RD, risk difference.

## Process outcomes and acceptability

For CC screening, 81.3% (*n/N* = 3,134/3,856) of the eligible women gave consent to the administrative check-up on their CC screening status, with 5.3% (*n/N* = 204/3,856) declining. The remainder either verbally affirmed having undergone timely screening (11.2%, *n/N* = 433/3,856) or passed by the research assistant without contact (2.2%, *n/N* = 85/3,856). Among women with overdue CC screening, 60.1% (*n/N* = 371/610) accepted to receive a vaginal self-sample, and 43.9% (*n/N* = 163/371) returned the self-sample within 6 months after the intervention.

For CRC screening, 76.4% (*n/N* = 3,787/4,955) of the eligible women gave consent to the administrative check-up on their screening status, while 5.3% (*n/N* = 261/4,955) declined. Among the remaining, 16.3% (*n/N* = 805/4,955) verbally affirmed having undergone timely screening, and 2.1% (*n/N* = 102/4,955) passed by. Among women overdue with CRC screening, 45.4% (*n/N* = 536/1,182) accepted to receive a new FIT, and 40.9% (*n/N* = 219/536) returned the screening within 6 months.

The questionnaire evaluating satisfaction with BC screening was answered by 66.2% (*n/N* = 3,688/5,567) of the women in the intervention group and 62.9% (*n/N* = 13,371/21,274) of the women in the control group. When comparing the positive answers (1–2) to the negative answers (3–4), more than 98% expressed satisfaction in each of the questionnaire items in both the intervention and the control group (Table 5). A statistically significant difference in satisfaction with the reception (Feeling welcome) was found in favour of the intervention units compared with the control units (*p* = 0.005).

The response rate to the questionnaire evaluating the intervention was 63.8% (*n/N* = 3,544/5,567), and the acceptability of the intervention was high with 87.4% (*n/N* = 2,994/3,426) indicating that they would accept a similar offer another time (Table 5). Detailed descriptive questionnaire data can be found in the Supporting information (S2–S4 Tables).

**Table 4. Participation in CC and CRC screening within 6 months of follow-up for women overdue for screening at baseline by screening history.**

| | Intervention group n/N % [95% CI] | Control group n/N % [95% CI] | RD Percentage points [95% CI] | P-value* |
|---|---|---|---|---|
| CC screening** | | | | |
| Screening participation | 195/610 32.0% [28.3%, 35.8%] | 149/2,424 6.1% [5.2%, 7.2%] | 25.8 [22.0, 29.6] | <0.001 |
| Unscreened | 69/247 27.9% [22.4%, 34.0%] | 40/1,036 3.9% [2.8%, 5.2%] | 24.1 [18.4, 30.0] | <0.001 |
| Under-screened | 126/363 34.7% [29.8%, 39.9%] | 109/1,388 7.9% [6.5%, 9.4%] | 26.9 [21.8, 32.0] | <0.001 |
| CRC screening*** | | | | |
| Screening participation | 281/1,182 23.8% [21.4%, 26.3%] | 396/4,466 8.9% [8.0%, 9.7%] | 14.9 [12.3, 17.5] | <0.001 |
| Unscreened | 137/698 19.7% [16.7%, 22.8%] | 194/2,610 7.4% [6.5%, 8.5%] | 12.2 [9.1, 15.3] | <0.001 |
| Under-screened | 144/484 29.8% [25.7%, 34.0%] | 202/1,856 10.9% [9.5%, 12.4%] | 18.9 [14.6, 23.2] | <0.001 |

n/N, number of women with overdue screening at baseline and screened within 6 months of follow-up/total number of women with overdue screening at baseline in the group.

*Comparison between the intervention and control groups, tested using a chi-squared test.

**Unscreened was defined as no cervical sample registered within the past 10 years; under-screened, as screened with a cervical sample at least once within the past 10 years but overdue at baseline.

***Unscreened was defined as no previous FIT despite an invitation; under-screened, as a minimum of 1 FIT but overdue at baseline.

95% CI, 95% confidence interval; CC, cervical cancer; CRC, colorectal cancer; FIT, faecal immunochemical test; RD, risk difference.

## Discussion

We performed a cluster-randomised crossover trial targeting women attending BC screening. The intervention resulted in a statistically significant absolute increase in total coverage in CC screening (4.8 percentage points) and in CRC screening (3.8 percentage points) compared with the control group receiving standard screening offers. Importantly, the increase in screening participation was considerable for both unscreened (CC screening: RD = 24.1 percentage points, CRC screening: RD = 12.2 percentage points) and under-screened women (CC screening: RD = 26.9 percentage points, CRC screening: RD = 18.9 percentage points). The intervention was feasible and highly accepted by the women.

To our knowledge, this is the first study directly using high coverage in one cancer screening programme to increase participation in others, and this was successful. Especially, among the unscreened women, 27.9% participated in CC screening and 19.7% participated in CRC screening. This is highly relevant, as unscreened women may benefit the most, given their elevated risk of having undetected disease or disease precursors. A high compliance to further clinical follow-up in the event of positive test results is also mandatory to derive benefits from the intervention. In our trial, almost all individuals with positive test results underwent relevant follow-up, consistent with recent Danish studies [35,36], showing a high adherence to triage after an HPV–positive CC screening and to colonoscopy after a FIT-positive CRC

**Table 5. Questionnaire data, satisfaction with BC screening and evaluation of the intervention.**

| | Intervention *n/N* (%) | Control *n/N* (%) | *P*-value* |
|---|---|---|---|
| Satisfaction with BC screening** | | | |
| Feeling welcome | 3,605/3,652 (98.7%) | 12,990/13,255 (98.0%) | 0.005 |
| Professionalism | 3,638/3,678 (98.9%) | 13,189/13,337 (98.9%) | 0.91 |
| Trust in examination | 3,626/3,684 (98.4%) | 13,150/13,349 (98.5%) | 0.71 |
| Overall satisfaction | 3,642/3,686 (98.8%) | 13,136/13,355 (98.4%) | 0.05 |
| Intention to participate next time*** | 3,499/3,510 (99.7%) | 12,773/12,825 (99.6%) | 0.44 |
| Evaluation of the intervention**** | | | |
| Oral information | 3,476/3,554 (97.8%) | .. | .. |
| Written information | 3,147/3,554 (88.6%) | .. | .. |
| Sufficient information | 3,422/3,554 (96.3%) | .. | .. |
| Meaningfulness | 2,974/3,554 (83.7%) | .. | .. |
| Combined screening | 2,552/3,554 (71.8%) | .. | .. |
| Would participate another time***** | 2,994/3,426 (87.4%) | .. | .. |

A translation of the questions is available in Supporting information (S1 Questionnaire).

*Comparison between the intervention and control groups, tested using a chi-squared test.

**Survey responses were given on a Likert scale ranging from 1 to 4. The reported values are the frequency and percentage of "1–2" (positive) Chi-squared tests compare "1–2" (positive) to "3–4" (negative).

***Survey responses were "yes" or "no." The chi-squared test compares "yes" to "no."

****Survey responses were given on a Likert scale ranging from 1 to 4 or "Do not know." The reported values are the frequency and percentage of "1–2" (positive).

*****Survey responses were "yes," "no," or "Do not know." The reported values are the frequency and % of "yes". Women indicating that this was not relevant for them were excluded (if they were no longer within the screening-eligible age range at the time of the next invitation or if they had a history of cancer or hysterectomy.

BC, breast cancer.

screening. All women aged 60 to 64 years who performed HPV re-testing 12 months after participating in the intervention with an initial HPV–negative self-sample also tested negative for HPV in the re-test. This suggests that these women may potentially be eligible to exit the screening programme after a single HPV–negative self-sample.

The reasons for women to participate following the intervention were not investigated in this study. However, since self-sampling was not a part of routine CC screening in the CDR at the time of the study, the availability of an alternative screening method could have been a significant factor for some participants. Nonetheless, self-sampling cannot explain the increase in CRC screening participation, as the CRC screening test offered during the intervention was the same as the one provided in the organised CRC screening programme. Hence, factors beyond the option of self-sampling contributed to the increase in screening participation. This is in line with results from a systematic review from 2022 [7] summarising factors favouring or hindering cancer screening participation at 3 operating levels: the individual level, the relational level, and the healthcare system level. We believe that our intervention succeeded in targeting all 3 levels at once. At the individual level, the offer of self-sampling for CC screening aimed at decreasing potential discomfort and negative emotions connected with a GP-collected cervical sample, and individual barriers such as difficulties in planning the appointment or forgetfulness are remedied. At the relational level, a face-to-face reminder, along with the opportunity to address concerns and receive additional information from healthcare staff, may provide practical and emotional support in both CC and CRC screening. In a meta-analysis from 2023, a face-to-face strategy was also shown to be the most effective to enhance participation in self-sampled CC screening [37]. Finally, the intervention targets the healthcare system level by addressing potential system barriers through facilitating easier access to screening [7].

It was a major strength of this intervention that it was embedded into the organised BC screening programme in an entire Danish region covering almost one-fourth of the Danish population. All 5 BC screening units in the region acted as intervention units and showcasing the feasibility of the intervention. This bodes well for the generalisability of the study findings to other Danish regions and to countries with comparable healthcare systems. Moreover, the trial provides a valid estimate for the predicted effect if the investigated approach were to be implemented. It is noteworthy that the marked increase occurred despite the already relatively high coverage compared to those seen in other countries [5]. Another strength was the high-quality data ensured by the Danish registers, with minimal missing data on baseline characteristics and data completeness in primary and secondary outcomes.

The design was crafted to optimise resource efficiency by leveraging the logistical advantages of incorporating 5 entire clusters, irrespective of patient recruitment. The utilisation of a crossover design aimed to diminish the likelihood of chance imbalances [27]. The crossover design was appropriate, given the intervention's ease of implementation and withdrawal, minimising consequential spill-over effects. Table 2 confirms the absence of differences between the intervention and control groups in important confounders such as demographics, socioeconomic status, and screening history. This supports our decision not to include a design effect due to cluster randomisation in both our sample size calculation and main analyses. Furthermore, the sensitivity analysis accounting for the clusters affirmed this decision. However, there may still be a risk of residual confounding entailed in our study design, but it is unlikely that this substantially influenced our results, giving the equal distribution of intervention and control clusters in the crossover design.

It is a limitation of the design that our trial was only powered to show a difference in coverage between the intervention and the control group. Thus, we cannot predict the preventive effect in terms of reduced risk of CC or CRC or the ability to facilitate CRC diagnosis at earlier stages. However, the increased participation among un- and under-screened women in particular holds promise for the effectiveness of this intervention in terms of achieving favourable long-term outcomes. It could even be speculated that the intervention may even serve to mitigate social inequality in cancer screening, since residents with low socioeconomic status are more likely not to participate under standard invitation procedures [38]. However, it is important to note that while the intervention addressed various common barriers to participation, including information, support, screening accessibility, and logistical challenges, other barriers such as cultural factors, language barriers, attitudes towards screening and socioeconomic status, were not directly addressed. Additionally, the study was not powered to show differences between different socioeconomic groups, and women attending BC screening may not represent the most vulnerable subpopulation of nonparticipants [39]. Finally, it is a limitation that our intervention solely focused on women attending breast cancer screening, even though men have been shown to exhibit the lowest participation in CRC screening [40] and face the highest risk of CRC [41], thus potentially having a greater need for intervention. Additionally, we lack information on whether any of the targeted residents were transgender. Addressing inequities, including those among socioeconomic groups and between genders require supplementary interventions for better targeting.

Regarding the general feasibility of the intervention, a potential limitation may be that the intervention in the study was conducted by research assistants rather than screening personnel at the BC units. Nevertheless, the intervention's design was adapted to ensure easy adoption by screening personnel, and future studies could explore potential challenges and benefits associated with integrating the approach into routine screening practices.

BC screening units were selected as the intervention location based on the participants' engagement in preventive care. This preexisting interest suggests that these women may be

more receptive to other health-related activities [23], and the feasibility established in this study may set the stage for easier access to multiple, simultaneous screenings including the opportunity to reach part of the underserved populations at low costs. For example, similar interventions could potentially be implemented in other healthcare settings, such as maternity visits for younger women or annual health examinations for the older population. The acceptability of such offers must be rigorously tested within the given study population.

It should be noted that after the trial, self-sampling has been integrated as an opt-in option along with the second reminder in CC screening in Denmark, which could potentially influence the intervention's effect in the CC screening programme. However, we believe that this will only be to a limited degree since self-sampling is not the only element of the intervention. Nevertheless, future trials comparing different self-sampling interventions such as different timings and settings for a self-sampling offer may contribute valuable insights. Additionally, employing qualitative methods could offer a more in-depth exploration of the underlying reasons.

In conclusion, based on the findings of this trial, the offer of face-to-face administrative check-ups on screening status for CC and CRC, combined with self-sampling screening offers for women overdue for these programmes, proved to be a feasible and effective approach during BC screening. This underscores the potential benefits of integrating combined screening offers within public health services, providing a pathway to enhance engagement in cancer screening programmes.

## Supporting information

**S1 Questionnaire. Questionnaires wording.**
(DOCX)

**S1 CONSORT Checklist. CONSORT Checklist for reporting randomised trials.**
(DOCX)

**S1 Table. Differences in coverage and participation between intervention and control groups in cervical and colorectal cancer screening accounting for clustering within screening units.**
(DOCX)

**S2 Table. Satisfaction with breast cancer screening (in the total study population, N = 17,059).**
(DOCX)

**S3 Table. Evaluation of the intervention (women in the total intervention group, who have responded to the questionnaire, N = 3,554).**
(DOCX)

**S4 Table. Evaluation of the intervention (women in the intervention group eligible for both screening programmes\*, who have responded to the questionnaire, N = 2,063).**
(DOCX)

## Acknowledgments

We thank all participants for taking part, the research assistants who executed the intervention (Marianne Rævsbæk Pedersen and Charlotte Riff), and the staff at the trial sites. Furthermore, we thank the Department of Pathology, Randers Regional Hospital, and the Department of

Clinical Biochemistry, Randers Regional Hospital for their collaboration. Roche Diagnostics sponsored HPV test kits for cervical cancer screening.

## Author Contributions

**Conceptualization:** Mette Bach Larsen, Lone Kjeld Petersen, Berit Andersen.

**Data curation:** Anne Dorte Lerche Helgestad, Mette Bach Larsen.

**Formal analysis:** Anne Dorte Lerche Helgestad, Mette Bach Larsen.

**Investigation:** Anne Dorte Lerche Helgestad.

**Methodology:** Anne Dorte Lerche Helgestad, Mette Bach Larsen, Sisse Njor, Mette Tranberg, Lone Kjeld Petersen, Berit Andersen.

**Project administration:** Anne Dorte Lerche Helgestad.

**Supervision:** Mette Bach Larsen, Berit Andersen.

**Validation:** Anne Dorte Lerche Helgestad, Mette Bach Larsen.

**Writing – original draft:** Anne Dorte Lerche Helgestad.

**Writing – review & editing:** Mette Bach Larsen, Sisse Njor, Mette Tranberg, Lone Kjeld Petersen, Berit Andersen.

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
