## [Editor Report · Decision Letter 0]

28 Feb 2024

Dear Dr Helgestad, 

Thank you for submitting your manuscript entitled "Increasing coverages in cervical and colorectal cancer screening using attendance for breast cancer screening as leverage (three birds with one stone): A cluster-randomised, crossover trial" for consideration by PLOS Medicine.

Your manuscript has now been evaluated by the PLOS Medicine editorial staff and I am writing to let you know that we would like to send your submission out for external peer review. 

Because your submission is a clinical trial, we ask that you provide a copy of your trial protocol as a supporting information file. By protocol, we mean the complete and detailed plan for the conduct and analysis of the trial that the ethics committee approved before the trial began. Please send this in the original language. If this is in a language other than English, please also provide a translation. Please detail any deviations from this study protocol in the Methods section of your manuscript. Your study protocol will be made available to the editors and reviewers.

Please re-submit your manuscript within two working days, i.e. by Mar 01 2024.

Feel free to email me at aschaefer@plos.org or us at plosmedicine@plos.org if you have any queries relating to your submission.

Kind regards,

Alexandra Schaefer, PhD

Associate Editor

PLOS Medicine

---

## [Decision Letter · Decision Letter 1]

16 Apr 2024

Dear Dr. Helgestad,

Thank you very much for submitting your manuscript "Increasing coverages in cervical and colorectal cancer screening using attendance for breast cancer screening as leverage (three birds with one stone): A cluster-randomised, crossover trial" (PMEDICINE-D-24-00636R1) for consideration at PLOS Medicine. 

Your paper was evaluated by an associate editor and discussed among all the editors here. It was also sent to independent reviewers, including a statistical reviewer. I apologize for the delay in providing you with a decision. The reviews are appended at the bottom of this email and any accompanying reviewer attachments can be seen via the link below:

[LINK]

In light of these reviews, I am afraid that we will not be able to accept the manuscript for publication in the journal in its current form, but we would like to consider a revised version that addresses the reviewers' and editors' comments. Obviously we cannot make any decision about publication until we have seen the revised manuscript and your response, and we plan to seek re-review by one or more of the reviewers. 

Please use the following link to submit the revised manuscript: https://www.editorialmanager.com/pmedicine/

We expect to receive your revised manuscript by May 07 2024. However, if this deadline is not feasible, please contact me by email, and we can discuss a suitable alternative.

Don't hesitate to contact me directly with any questions (aschaefer@plos.org). If you reply directly to this message, please be sure to 'Reply All' so your message comes directly to my inbox.

We look forward to receiving your revised manuscript. 

Sincerely,

Alexandra Schaefer, PhD

PLOS Medicine

plosmedicine.org

EDITORIAL COMMENTS

We agree with the reviewers that more details should be provided on screening rates and screening programs in Denmark. We also ask you to discuss the potential for selection bias, as individuals who attend screening for one reason are more likely to accept screening.

***Please note: not all will apply to your paper, but please check each item carefully

GENERAL COMMENTS

1) Please cite the reference numbers in square brackets. Citations should be preceding punctuation.

COMPETING INTEREST

All authors must declare their relevant competing interests per the PLOS policy, which can be seen here: https://journals.plos.org/plosmedicine/s/competing-interests

For authors with ties to industry, please indicate whether any of the interests has a financial stake in the results of the current study.

DATA AVAILABILITY STATEMENT 

The Data Availability Statement (DAS) requires revision. For each data source used in your study: 

TITLE

Please revise your title according to PLOS Medicine's style. Your title must be nondeclarative and not a question. It should begin with main concept if possible. "Effect of" should be used only if causality can be inferred, i.e., for an RCT. Please place the study design ("A randomized controlled trial," "A retrospective study," "A modelling study," etc.) in the subtitle (ie, after a colon).

ABSTRACT

1) Please report your abstract according to CONSORT for abstracts, following the PLOS Medicine abstract structure (Background, Methods and Findings, Conclusions). https://www.equator-network.org/reporting-guidelines/consort-abstracts/

2) PLOS Medicine requests that main results are quantified with 95% CIs as well as p values. When reporting p values please report as p<0.001 and where higher as the exact p value p=0.002, for example. For the purposes of transparent data reporting, if not including the aforementioned please clearly state the reasons why not. When a p value is given, please specify the statistical test used to determine it.

3) Throughout, suggest reporting statistical information as follows to improve clarity for the reader “22% (95% CI [13%,28%]; p</=)”. Please be sure to define all numerical values at first use. Please amend throughout the abstract and main manuscript. Please note the use of commas to separate upper and lower bounds, as opposed to hyphens as these can be confused with reporting of negative values.

3) Please ensure that all numbers presented in the abstract are present and identical to numbers presented in the main manuscript text.

4) Please include the study design, population and setting, number of participants, years during which the study took place (enrollment and follow up), length of follow up, and main outcome measures.

5) Please specify who was blinded to the intervention and control, define the intervention and control states, provide the number in each group, state that analysis was intention to treat and provide the number of participants lost to follow up in each group.

6) Please include the actual amounts and/or absolute risk(s) of relevant outcomes (including NNT or NNH where appropriate), not just relative risks or correlation coefficients. (example for absolute risks: PMID: 28399126).

7) Please include the important dependent variables that are adjusted for in the analyses.

8) Please define all abbreviations including those for statistical reporting at first use.

9) In the last sentence of the Abstract Methods and Findings section, please describe the main limitation(s) of the study's methodology.

10) Please include the clinical trial registry number in the abstract.

AUTHOR SUMMARY

At this stage, we ask that you include a short, non-technical Author Summary of your research to make findings accessible to a wide audience that includes both scientists and non-scientists. The Author Summary should immediately follow the Abstract in your revised manuscript. This text is subject to editorial change and should be distinct from the scientific abstract. Ideally each sub-heading should contain 2-3 single sentence, concise bullet points containing the most salient points from your study. In the final bullet point of ‘What Do These Findings Mean?’, please include the main limitations of the study in non-technical language. Please see our author guidelines for more information: https://journals.plos.org/plosmedicine/s/revising-your-manuscript#loc-author-summary

METHODS AND RESULTS

1) PLOS Medicine requests that main results are quantified with 95% CIs as well as p values. We suggest reporting statistical information as detailed above – see under ABSTRACT

2) Please present numerators and denominators for percentages (at least in the Tables [not necessarily each time they're mentioned]).

3) Please complete the CONSORT checklist and ensure that all components of CONSORT are present in the manuscript, including how randomization was performed, allocation concealment, blinding of intervention, definition of lost to follow-up, power statement. When completing the checklist, please use section and paragraph numbers, rather than page numbers.

4) Please include the study protocol document and analysis plan, with any amendments, as Supporting Information to be published with the manuscript if accepted. Please note that changes in the analysis-- including those made in response to peer review comments-- should be identified as such in the Methods section of the paper, with rationale.

5) Please present the safety data for the study including numbers of specific events and whether or not adverse events are thought to be related to the intervention.

DISCUSSION

Please present and organize the Discussion as follows: a short, clear summary of the article's findings; what the study adds to existing research and where and why the results may differ from previous research; strengths and limitations of the study; implications and next steps for research, clinical practice, and/or public policy; one-paragraph conclusion (no subheading).

FIGURES AND TABLES 

1) Please provide titles and legends for all figures and tables (including those in Supporting Information files). 

2) Please define all abbreviations used in each figure/table (including those in Supporting Information files). 

3) Please consider avoiding the use of red and green in order to make your figure more accessible to those with color blindness. 

SUPPLEMENTARY MATERIAL

1) For supplementary figures and tables, please see the general comments under TABLES and FIGURES and amend accordingly.

2) We suggest reporting statistical information as detailed above – see under ABSTRACT. Please be sure to define all numerical values.

3) As for the main manuscript, please indicate whether analyses are adjusted to help facilitate transparent data reporting please also detail the factors adjusted for and present the unadjusted analyses for comparison. If not, please clearly state the reasons why not.

4) Please cite your Supporting Information as outlined here: https://journals.plos.org/plosmedicine/s/supporting-information

REFERENCES

1) PLOS uses the numbered citation (citation-sequence) method and first six authors, et al.

2) Please ensure that journal name abbreviations match those found in the National Center for Biotechnology Information (NCBI) databases (http://www.ncbi.nlm.nih.gov/nlmcatalog/journals), and are appropriately formatted and capitalised.

3) Where website addresses are cited, please specify the date of access (e.g. [accessed: 16/09/2023]).

4) Please also see https://journals.plos.org/plosmedicine/s/submission-guidelines#loc-references for further details on reference formatting. 

Comments from the reviewers:

Reviewer #1: excellent project. excellent methods and analysis. 

Reviewer #2: The authors present results from an elegant cluster randomised crossover trial of offering three cancer screening tests at one clinical setting to test whether it improves uptake and overall coverage.

The title of the manuscript uses common parlance of "three birds with one stone" which I find unacceptable in a scientific report. The authors must replace this with a more suitable medically descriptive title. In addition I have noted further issues as detailed below.

Issues identified:

1. Authors state no funding received for this work but then mention under competing interests that the cobas 4800 HPV tests were sponsored by Roche. Is this not funding? I find it hard to believe there was no specific funding allocated for the trial. Was there not institute funding for the staff and /or student time for conducting the trial?

2. There are a few English readability errors in the abstract, such as on line 29: "Screening participation (in Denmark) remain(s) suboptimal in …."

3. It would be more correct to state "total screening coverage for the three programmes" than "overall coverage" in the abstract, as well as in the body of the text. 

4. The introduction, from lines 63 to 73 makes the case that the main reason for non-participation in screening is "temporary impediments or forgetfulness". The discussion lines 436-437 also speaks of these "individual barriers". In my experience of cancer screening programme research, this is an impressively large oversimplification. It is a significant weakness of this study that it does not consider equity, access, and socioeconomic factors as barriers to screening. These factors are raised by the authors, for the first time, in the discussion on lines 449 to 461, although not as an admission of a study weakness which must be addressed.

5. Outcomes, line 166 onward: Terminology could be improved. Difference between the control and intervention group with respect to CC and or CRC screening (uptake) six months after the (offer of self-testing) for women who were overdue for CC/CRC screening at the intervention date. 

6. In your methods: 

a. You need a section in the methods briefly describing the ethics approval for the study and the process of informed consent undertaken for participants.

b. It is not clear that HPV self sampling is not already implemented in your programme, this is only mentioned in the discussion in lines 426-427. There is also no information on the method of your self sampling test (cobas 4800), whether it is validated for self sampling in your setting and the self sampling device used. Please update your methods with a section on all this information.

c. It is not clear until the discussion that the self test kits were mailed out to participants, after the breast screening accession, and that women weren't able to do the test onsite.

d. Method: how are participants managed clinically? Do you not have an algorithm for directing those who are HPV16/18 straight to colposcopy? With those positive for non-16/18 HPV types given cytology and triage? Brief information on this would be helpful. 

e. Method: Study exclusion criteria, did you not have a category for loss to follow-up due to non-contactability? Or was this not an issue? 

7. Results: 

a. Table 5 the section on "satisfaction with breast cancer screening" it is not clear what this data relates to. Is it the 1-2 positive responses on the Likert scale, as for the "Evaluation of the intervention"? Please correct the legend associated with this figure

b. Table 4. Was there any statistical difference between the groups? I.e. in the CC groups, was there any difference between uptake in the unscreened and underscreened groups? Similarly for the CRC groups, was there any difference between unscreened and underscreened groups?

c. Table 2. I note that some columns do not add up to the total N due to missing data, I would prefer that you add a line to the ethnicity, marital status, and education rows for "data not given" to enable easy viewing of the impact of this on the results. By my calculations, it ranges from 4 for the CC intervention groups to 146 in the control group for education. 

8. Discussion: In addition to addressing the weakness of your study I addressed in point 4, the statement and reference on line 460 that "offering self-sampling at GPs has shown promising results" should be removed as it is is out of date. Self sampling for Cervical cancer screening is now implemented in GP clinics, via mailout and many other settings in many countries, including the Netherlands (since 2017), Australia and New Zealand. Indeed, the metaanalysis by Costa et al (2023) that is cited in the discussion, found that "overall screening participation was higher among women invited for s

---

## [Decision Letter · Decision Letter 2]

31 May 2024

Dear Dr. Helgestad,

Thank you very much for re-submitting your manuscript "Increasing coverage in cervical and colorectal cancer screening by leveraging attendance at breast cancer screening: A cluster-randomised, crossover trial" (PMEDICINE-D-24-00636R2) for review by PLOS Medicine.

Thank you for your detailed response to the editors' and reviewers' comments. I have discussed the paper with my colleagues and the academic editor, and it has also been seen again by the statistical reviewer. The changes made to the paper were satisfactory to the reviewer. As such, we intend to accept the paper for publication, pending your attention to the editorial comments below in a further revision. When submitting your revised paper, please once again include a detailed point-by-point response to the editorial comments.

[LINK]

In revising the manuscript for further consideration here, please ensure you address the specific points made by each reviewer and the editors. In your rebuttal letter you should indicate your response to the reviewers' and editors' comments and the changes you have made in the manuscript. Please submit a clean version of the paper as the main article file. A version with changes marked must also be uploaded as a marked up manuscript file. Please also check the guidelines for revised papers at http://journals.plos.org/plosmedicine/s/revising-your-manuscript for any that apply to your paper. 

We ask that you submit your revision within 1 week (Jun 07 2024). However, if this deadline is not feasible, please contact me by email, and we can discuss a suitable alternative.

Please do not hesitate to contact me directly with any questions (atosun@plos.org). If you reply directly to this message, please be sure to 'Reply All' so your message comes directly to my inbox.

We look forward to receiving the revised manuscript.   

Sincerely,

Alexandra Tosun, PhD

Associate Editor 

PLOS Medicine

plosmedicine.org

Requests from Editors:

GENERAL COMMENTS

We encourage you to include the original protocol document and analysis plan, with any amendments, as Supporting Information to be published with the manuscript if accepted (along with the published version of the protocol).

Under data availability you have stated that “The participants were not asked to give consent for publication of the questionnaire data.” whereas in lines 296-298, you have stated that informed consent was not required. Please comment and update the details with appropriate information. 

Throughout the manuscript, please provide more detailed references to the results in the Supplementary Material (e.g. “S1 Table 2”).

We appreciate your response to the comment by Reviewer #4 regarding the inclusion of individuals who identify as gender non-binary and transgender men with a cervix. We encourage you to include these details in the Methods section and also in the discussion as appropriate.

ABSTRACT

1) l.36: Please describe briefly what “control” entails, for example, as done in lines 192-193.

2) l.38: Please change to: “…and women with overdue screening were offered self-sampling.” or similar (please do not use the term “overdue women”). Please amend throughout the abstract and the main text.

3) l.42: Please introduce the abbreviation for confidence intervals here (“…with 95% confidence intervals (CI).”)

4) l.54: Please remove the word “remarkable”. 

5) l.56: Please correct the trial registration number to “NCT05022511”.

AUTHOR SUMMARY

l.75: Please change to: “We conducted…”

INTRODUCTION

1) l.109: Please change “currently” to an exact time detail, e.g. “as of 2024”.

2) l.115: When stating age, please add the relevant unit, e.g. “aged 23-64 years”. Please revise throughout the main text.

METHODS AND RESULTS

1) l.161: “The rationale and design of this trial have been described in details elsewhere [25].”

2) l.212 “Outcomes”: The protocol lists the outcomes of [incidence of HPV-positive cases in women 60–64 years after 12 months with an initial negative HPV sample]. (a) Can you please present those results as part of this manuscript, or indicate why that is not possible? (b) If you do not wish to include these results in the current paper, can you please indicate when you plan to publish these results?

3) ll.231-233: The following outcomes measures, “Secondary outcomes also included the proportion of women eligible for all three programmes, the proportion of these women overdue for both CC and CRC screening, and their participation in both screenings within the follow-up period.”, appear to differ between the submitted manuscript and the protocol. Please clarify and explain the discrepancy. If the outcomes were not prespecified in the protocol, please indicate that they were post hoc and explain why they were added. Post hoc comparisons should be presented as hypothesis generating rather than conclusive.

4) Figure 2: Please add a footnote for the asterisk.

5) Table 2: Regarding missing data, you stated that only descriptive data for socio-demographic variables were missing for a very small proportion of the study population. We understand that there may be limitations in presenting the data due to Danish data protection legislation. However, we wonder whether adding a row to report 'missing data' for each variable in Table 2 will identify a person. In any case, we have noticed that the calculated percentages are incorrect, as it appears that you have used denominators for the calculations that exclude those with missing data. Please revise and use the total N to calculate the percentages (which may then not equal 100%).

6) Table 2: Please add a definition for “Western immigrants” and “Non-western immigrants” and exchange the word “immigrant” with “migrant”. Also, please define “FIT” at first use in the footnote.

7) Table 3: Please change “[82.9%, 84.1%9” to “[82.9%, 84.1%]” (i.e., missing end bracket).

8) Table 4: Please define “FIT” at first use in the footnote.

9) l.371 ff: Please add a denominator when presenting results (e.g. “..was 8.0% (n/N=13/163; 95% CI [4.3%, 13.3%])”). Please revise throughout the main text.

10) ll.375-377: Could you please comment on the restrictions based on Danish data protection legislation? It seems that in some cases it is okay to provide small numbers, and in others not. Please check if it is possible to provide an exact number, as the number alone is unlikely to be a personal identifier.

11) ll.383 ff: Please present either the n number (along with a denominator) first, followed by the percentage in parentheses, or the other way around. Please avoid mixing. For example: “For CC screening, 81.3% (n=3,134) of the eligible women gave consent to the administrative check up on their CC screening status, with 5.3% (n=204) declining.”

12) l.398: Please report the exact p-value.

13) Table 5: Please include the denominators for the different groups and check that the percentages were calculated appropriately. “Women indicating that this was not relevant for them were excluded.” – We feel that all respondents should be included; please explain why these women were excluded. 

DISCUSSION

1) We feel that the discussion is currently a bit disjointed. For example, the limitations of the study are not presented in a dedicated paragraph but are interspersed between other paragraphs. We would encourage you to revise the discussion to improve clarity and structure; typically, limitations are discussed in a standalone paragraph.

2) ll.428-429/l.514: Please temper claims of primacy of results by stating, "to our knowledge" or something similar.

3) ll.437-443: We feel that the message this paragraph is intended to convey is currently unclear. Please revise for clarity.

4) l.462: “This bodes well for the generalisability of the study findings” – We feel that this statement requires a bit more context, as the results of the study could be generalized to Denmark as a whole or to countries with similar health care systems.

5) l.527: Please remove the “Conclusion” subheading.

SUPPLEMENTARY MATERIAL

Thank you for providing the CONSORT checklist. Please replace the page numbers with paragraph numbers per section (e.g. "Methods, paragraph 1"), since the page numbers of the final published paper may be different from the page numbers in the current manuscript. Also, please add the following statement, or similar, to the Methods: "This study is reported as per the Consolidated Standards of Reporting Trials (CONSORT) guideline for cluster randomised trials (S1 Checklist)."

REFERENCES

Where website addresses are cited, please specify the date of access including day, month and year (e.g. [accessed: 16/09/2023]).

SOCIAL MEDIA

To help us extend the reach of your research, please provide any X (formerly known as Twitter) handle(s) that would be appropriate to tag, including your own, your co-authors’, your institution, funder, or lab. Please enter in the submission form any handles you wish to be included when we post about this paper.

Comments from Reviewers:

Reviewer #3: Thank you to the authors for addressing all my previous comments well. I have no further issues to raise.

[LINK]

General Editorial Requests

---

## [Editor Report · Decision Letter 3]

10 Jun 2024

Dear Dr. Helgestad,

Thank you very much for re-submitting your manuscript "Increasing coverage in cervical and colorectal cancer screening by leveraging attendance at breast cancer screening: A cluster-randomised, crossover trial" (PMEDICINE-D-24-00636R3) for review by PLOS Medicine.

I have discussed the paper with my colleagues and there remain a few outstanding requests which need to be addressed prior to publication.

[LINK]

We look forward to receiving the revised manuscript by Jun 17 2024 11:59PM.   

Kind regards,

Pippa

Philippa Dodd MBBS MRCP PhD

Senior Editor 

PLOS Medicine

plosmedicine.org

pdodd@plos.org

(on behalf of Alexandra Tosun, PhD)

Requests from Editors:

1) COMPETING INTERESTS STATEMENT 

please replace ‘proces’ with ‘process’.

2) ABSTRACT

In the last sentence of the methods and findings section please briefly detail the main limitations of the study’s methodology.

3) AUTHOR SUMMARY

Please swap bullet points at lines 83 and 85 to ensure bullet point at line 85 is the first bullet point. Please also revise line 85 to read, ‘This study suggests that combining preventive screening services…’

Line 86 – it isn't entirely clear what i meant by 'health offers'. Suggest, ‘…public health interventions…’ instead perhaps?

Line 87 – suggest, ‘…and is likely to require a distinct approach…’

In the final bullet point of the ‘What do these findings mean sub-section’ please include the main limitations of the study in non-technical language.

4) METHODS AND RESULTS 

Line 161: “The rationale and design of this trial have been described in details elsewhere [25].” Please replace ‘details’ with ‘detail’.

5) TABLE 2 FOOTNOTE 

For explicit clarity suggest, “Exact numbers are not given in pursuance of Danish data protection legislation, which does not permit reporting of individual data where total numbers are lower than 3.”

Line 330 – please revise ‘**Western immigrants’ to read ‘**Western migrants’. Please check carefully for the same elsewhere and amend throughout all subsections of the main manuscript and supporting files as necessary.

6) TABLE 5 

Line numbers are overwritten on the final column of the table, please amend.

7) STATEMENTS OF DECLARATION

Lines 560 and 567 – please remove these statements from the end of the discussion and include only in the manuscript submission form. They will be compiled as metadata at the time of publication.

8) SUPPORTING INFORMATION

Please label ‘S1 File supplementary material’ as ‘S1 Appendix’.

Within the S1 Appendix please label tables as ‘S1 Table, S2 Table’ etc in numbered sequence and similarly, figures as ‘S1 Fig, S2 Fig’.

Please label ‘S2 File CONSORT’ as ‘S2 CONSORT Checklist’.

Please be reminded to amend citations to the supporting information files/figures/tables throughout all sections of the manuscript (and supporting files).

Please be reminded to amend the titles/captions at line 699 onwards.

9) STUDY PROTOCOL 

There seems to be some confusion regarding the required study protocol. A reference and a citation to the published BMJ article will suffice, we do not need to re-publish this item. 

Please remove the PDF file of the BMJ article and instead upload a copy of the protocol document which would have been submitted as part of the trial approval process. The BMJ article was published sometime after the trial commenced thus cannot be the ‘original’ protocol (it is the published version of the original protocol). 

Please label the uploaded protocol document as ‘S3 Protocol’.

[LINK]

---

## [Editor Report · Decision Letter 4]

20 Jun 2024

Dear Dr Helgestad, 

On behalf of my colleagues and the Academic Editor, Wei Zheng, I am pleased to inform you that we have agreed to publish your manuscript "Increasing coverage in cervical and colorectal cancer screening by leveraging attendance at breast cancer screening: A cluster-randomised, crossover trial" (PMEDICINE-D-24-00636R4) in PLOS Medicine.

I appreciate your thorough responses to the reviewers' and editors' comments throughout the editorial process. We look forward to publishing your manuscript, and editorially there are only a few remaining minor stylistic/presentation points that should be addressed prior to publication. We will carefully check whether the changes have been made. If you have any questions or concerns regarding these final requests, please feel free to contact me at atosun@plos.org.

Please see below the minor points that we request you respond to:

1) Competing Interests statement: The word ‘proces’ is still written incorrectly in the statement - please replace with ‘process’.

2) References [25], [40], [41]: Please change the format to six authors et al. 

3) Thank you for your explanation regarding the study protocol. Please remove the BMJ article from the file inventory. A reference and citation to the published BMJ article is sufficient, we do not need to republish the article. For the sake of transparency, we ask you to include a brief explanation of the lack of a study protocol (prior to the start of the trial) in the Methods, including details of the Danish regulations if you wish.

PRESS

Sincerely, 

Alexandra Tosun, PhD 

Associate Editor 

PLOS Medicine